# Functionalized GelMA/CMCS Composite Hydrogel Incorporating Magnesium Phosphate Cement for Bone Regeneration

**DOI:** 10.3390/biomedicines13020257

**Published:** 2025-01-21

**Authors:** Xingyu Wang, Xiping Zhang, Changtian Gong, Jian Yang, Jingteng Chen, Weichun Guo

**Affiliations:** Department of Orthopedics, Renmin Hospital of Wuhan University, Wuhan 430060, China; 2017302180019@whu.edu.cn (X.W.); 2016302180026@whu.edu.cn (X.Z.); gongct@whu.edu.cn (C.G.); yangjian1986@whu.edu.cn (J.Y.); jtchen@whu.edu.cn (J.C.)

**Keywords:** GelMA hydrogel, bone regeneration, MPC magnesium, carboxymethyl chitosan

## Abstract

**Background:** Bone regeneration remains a challenging issue in tissue engineering. The use of hydrogels as scaffolds for bone tissue repair has gained attention due to their biocompatibility and ability to mimic the extracellular matrix. This study aims to develop a functionalized GelMA/CMCS composite hydrogel incorporating magnesium phosphate cement (MPC) for enhanced bone regeneration. **Methods:** These composites were developed by incorporating potassium magnesium phosphate hexahydrate (KMgPO_4_·6H_2_O, MPC) powders into methacrylated gelatin/carboxymethyl chitosan (GelMA-C) hydrogels. The material’s mechanical properties, antibacterial performance, and cytocompatibility were evaluated. In vitro experiments involved cell viability and osteogenic differentiation assays using rBMSCs as well as angiogenic potential assays using HUVECs. The hydrogel was also assessed for its potential in promoting bone repair in a rat (Sprague-Dawley) model of bone defect. **Results:** The developed GelMA-CM composite demonstrated improved mechanical properties, biocompatibility, and osteogenic potential compared to individual GelMA or CMCS hydrogels. Incorporation of MPC facilitated the sustained release of ions which promoted osteogenic differentiation of pre-osteoblasts. In vivo results indicated accelerated bone healing in the rat bone defect model. **Conclusions:** The functionalized GelMA-CM composite could be a viable candidate for clinical applications in bone regeneration therapies.

## 1. Introduction

Bone defects from trauma, osteoporosis, infection debridement, and tumor resection significantly strain global public healthcare systems [1]. The global prevalence of osteoporosis and osteopenia is 9.7% (95%CI, 18.0–21.4%) and 40.4% (95%CI, 36.9–43.8%), respectively. The challenge of restoring normal bone morphology and function using biomaterials remains largely unmet [2]. One major issue is the design of scaffolds that can adequately mimic the native bone structure. Despite progress, many scaffold materials still suffer from limitations in their mechanical strength, bioactivity, and ability to support vascularization. These materials often lack the necessary properties to support long-term bone regeneration and integration with host tissues. Another challenge lies in achieving optimal cellular integration. While scaffolds provide the physical structure, ensuring effective cell adhesion, proliferation, and differentiation within these materials remains difficult. The complex interactions between cells and scaffolds, as well as the difficulty in replicating the native bone microenvironment, continue to hinder successful bone regeneration. Vascularization is also a critical issue in bone tissue engineering. While small tissue constructs can survive without blood supply, larger bone grafts require an adequate blood vessel network to deliver nutrients and oxygen for cell survival. Inducing sufficient vascularization within engineered tissues is thus a major hurdle in creating large-scale bone constructs. Additionally, immunological responses and biocompatibility concerns remain prevalent in tissue engineering. The body’s immune response to implanted materials can lead to inflammation and rejection, which complicates the success of tissue integration. Therefore, the development of scaffolds that are both biocompatible and capable of supporting immune tolerance is essential for advancing bone tissue engineering applications.

Bone tissue is an organic–inorganic composite composed of minerals and collagen. For an effective bone repair material, several key properties must be carefully considered. Biocompatibility is essential, as the material must seamlessly integrate with bone tissue without triggering an immune response, thereby promoting cell adhesion, proliferation, and differentiation. Osteoconductivity is equally critical, as the material should facilitate bone cell migration and the formation of new bone tissue through an optimized porous structure that supports vascularization. In certain cases, osteoinductivity plays a crucial role, where the material induces the differentiation of stem cells into osteoblasts, further enhancing bone formation. Additionally, the mechanical strength of the material must closely resemble that of natural bone, especially in load-bearing regions, to withstand physiological stresses without failure. The degradation rate of the material must be synchronized with the rate of bone regeneration, ensuring that the scaffold maintains its structural integrity long enough to support tissue formation, while gradually degrading in a controlled manner. Therefore, creating functionalized composite materials is essential for furthering the field of bone tissue engineering.

Hydrogels derived from natural polymers, including gelatin, collagen, dextran, and silk fibroin, mimic the extracellular matrix (ECM) and provide a three-dimensional scaffold that supports cell attachment and growth [3]. Among various biopolymers, gelatin, derived from the hydrolysis of collagen, the primary component of the ECM, has been extensively utilized in tissue engineering due to its antigenicity and biocompatibility. Additionally, the degradation rate of natural hydrogels is often difficult to control precisely, which may lead to premature degradation before adequate tissue regeneration, or slower degradation, potentially hindering integration with surrounding tissues. Chitosan (CS), a partially deacetylated derivative of chitin, resembles glycosaminoglycans (GAGs), another crucial ECM component, making it an ideal scaffold material suited for use in tissue engineering applications [4]. Among the most prevalent chitosan derivatives, carboxymethyl chitosan (CMC) is notable for its biocompatibility, biodegradability, and osteogenic properties. Furthermore, CS displays significant antimicrobial and hemostatic properties, attributed to the cationic nature of its amino groups [5,6]. Magnesium, which constitutes approximately 67% of the bone tissue’s mineral content, is the fourth most prevalent metal in the human body [7]. Magnesium ions (Mg^2+^) are essential participants in more than 600 enzymatic processes, including those involved in energy metabolism and protein synthesis [8]. Mg^2+^ is integrated into the bone mineral lattice, contributing to the regulation of natural bone mechanics [9].

Despite the advantages of gelatin- and chitosan (CS)-based hydrogels, their poor mechanical properties significantly hinder their application as scaffolds, particularly in load-bearing tissues [10,11]. Common strategies to enhance the mechanical integrity of hydrogels include increasing crosslink density, reducing the degree of gel swelling, fabricating interpenetrating networks, and incorporating fiber-reinforcing agents [12,13,14,15]. Although recent advancements have introduced alternative strengthening mechanisms, such as topological hydrogels, tetra-PEG hydrogels, and nanocomposite hydrogels, double-network (DN) hydrogels remain the most prevalent due to their exceptional mechanical strength [14,16,17,18]. Hairui Suo et al. demonstrated that the interpenetration of a photo-crosslinked gelatin network with a hydrophobic CS network effectively enhances mechanical properties [19]. Similarly, Hong Chen et al. showed that oxidized dextran can interact with gelatin through a Schiff base reaction, resulting in an OD/gel hydrogel with a three-dimensional network structure [20]. Based on these findings, we hypothesized that carboxymethyl chitosan (CMCS) can react with GelMA via a Schiff base reaction to form a three-dimensional GelMA-C hydrogel network, with MPC powder serving as an inorganic filler to further enhance the mechanical properties of the composite.

In this work, we performed an in-depth evaluation of the physical and chemical characteristics of GelMA, GelMA-C, and GelMA-CM hydrogels. Furthermore, their biocompatibility and ability to promote bone regeneration were assessed via both in vitro and in vivo experiments.

## 2. Materials and Methods

### 2.1. Materials

In this study, all reagents used were of analytical grade. Potassium dihydrogen phosphate (KH_2_PO_4_) and light magnesium oxide (MgO) powders, each with a purity of ≥98%, were sourced from Sinopharm Group Co., Ltd. (Shanghai, China). Carboxymethyl chitosan (CMCS) (molecular weight: 50 kDa; carboxylation degree: 83.5%) was sourced from Zhejiang Aoxing Biotechnology. GelMA (EFL-GM-60, amino substitution degree: 60 ± 5%) was acquired from Engineering For Life Co., Ltd., in Suzhou and stored at −20 °C.

### 2.2. Synthesis of GelMA-CM

GelMA was synthesized according to the method reported previously [21]. The magnesium oxide (MgO) and potassium dihydrogen phosphate (KH_2_PO_4_) powders were ground in a ball mill (F-P400, Focucy, Changsha City, China) and then sieved through a 200-mesh screen to achieve particles with a diameter of approximately 75 μm. The ground MgO and KH_2_PO_4_ powders were mixed in a molar ratio of 1.5:1 with deionized water at a concentration of 2 g/mL. Following self-setting and demolding, the potassium magnesium phosphate hexahydrate (KMgPO_4_·6H_2_O, MPC) powders were obtained by further grinding and sieving through a 200-mesh screen for 2 h. At room temperature, the photoinitiator, 5% MPC powder, and 5% carboxymethyl chitosan (CMCS) powder were incorporated into GelMA based on a weight-to-volume ratio (g/mL). The photoinitiator, 2-hydroxy-4′-(2-hydroxyethoxy)-2-methylphenylacetone (Irgacure 2959), was used at a concentration of 0.5% (*w*/*v*). The resulting pre-gel solution was placed in a water bath at 60–70 °C until fully dissolved (Figure 1). The mixture was then cast into a circular mold with a diameter of 6 mm and a thickness of 1 mm and subjected to UV light for crosslinking. Based on their composition, the resulting composites were designated as GelMA, GelMA-C, and GelMA-CM.

### 2.3. Characterization of GelMA-CM

The composites were freeze dried using a vacuum freeze dryer for 72 h, after which they were ground into powder. X-ray diffraction (XRD) analysis was performed to characterize the GelMA-CM composition over a 2θ range of 10–65°, using Cu-Kα radiation with a voltage of 40 kV and a current of 50 mA (XPert Pro, PANalytical B.V., Almemo, The Netherlands). Fourier-transform infrared spectroscopy (FT-IR) was used to analyze chemical bonds using a Nicolet 6700 instrument (Thermo Electron Scientific Instruments, Thermo Fisher Scientific, Waltham, MA, USA). The samples, mixed with KBr powder, were pressed into tablets, and the infrared spectra were recorded in the range of 4000 to 500 cm^−1^. SEM was used to examine the surface morphology of the samples using a Zeiss SIGMA instrument (Carl Zeiss, Cambridge, UK).

### 2.4. Mechanical Analysis and Mg^2+^ Release of GelMA-CM

Using a rotating rheometer with an 8 mm diameter plate–plate geometry and a 1 mm gap, the viscoelastic behavior of various GelMA hydrogel samples, prepared in disk form, was assessed. The loss modulus (G″) and storage modulus (G′) for each sample were determined as a function of frequency, with a constant strain of 0.1 N applied and the frequency varied from 0.01 to 10 Hz. The measurements were conducted at 25 °C. The following values for the storage modulus (G′), loss modulus (G″), and loss tangent (tan δ) were obtained.$$G^{\prime }=\frac{\sigma_{0}}{\varepsilon_{0}}\cos\delta ,G^{\prime \prime }=\frac{\sigma_{0}}{\varepsilon_{0}}\sin\delta ,\tan\delta =\frac{G^{\prime \prime }}{G^{\prime }}$$

The prepared GelMA-CM composite was placed in a 6-well plate and immersed in 5 mL of phosphate-buffered saline (PBS). The concentration of magnesium ions (Mg^2^⁺) was measured and recorded on days 1, 3, 5, 7, 14, 21, and 28. For each time point, 2 mL of the sample was withdrawn, diluted to 10 mL with PBS, and analyzed using inductively coupled plasma optical emission spectrometry (ICP-OES; PerkinElmer, Waltham, MA, USA, Optima 7000DV). The Mg^2^⁺ concentration in the solution was determined by comparing it to standard Mg^2^⁺ solutions. Three parallel measurements were performed to obtain an average value. To continue the release process, 2 mL of fresh PBS was added, and the system was maintained at 37 ± 1 °C for further incubation.

### 2.5. In Vitro Studies

#### 2.5.1. Cellular Proliferation

After sterilization, the samples were placed in 24-well plates, and 1 × 10^4^ SD rBMSCs (from Servior) were seeded onto each sample. They were cultured at 37 °C in a 5% CO_2_ incubator using rat bone marrow mesenchymal stem cell complete medium. The culture medium was replaced daily. At specified time points (days 1, 3, and 7), the samples were washed with DPBS and treated with EZ-Cytox diluent for 90 min. Absorbance values at 450 nm were used to compare viability between groups. Triplicates of all experiments were performed.

#### 2.5.2. Cell Morphology

Following sterilization, the samples were placed in individual wells of 6-well plates, where 5 × 10^5^ HUVECs were seeded onto each sample. After 24 h, the cells were washed three times with PBS and then stained with Fluorescein Diacetate (FDA, D1219A, MeilunBio, Dalian, China) and Propidium Iodide (PI, KGA107, KeyGEN, Nanjing, China). The live and dead cells were examined under an inverted microscope (IX71, Olympus, Tokyo, Japan). After this incubation, the cells were fixed in 4% paraformaldehyde for 15 min following three PBS washes. Actin microfilaments were stained for 5 min with SF488 phalloidin (Solarbio, Wuhan, China), while the nuclei were stained with DAPI for 10 min. The arrangement of the cytoskeleton was then visualized using a fluorescence microscope (DM750, Leica, Shanghai, China).

Cell adhesion ability was assessed using microscopy imaging. Image processing and analysis were performed using ImageJ software (1.53e). The images were first thresholded to separate the cells from the background, and the cell areas were calculated. The adhesion area of the cells was used to evaluate the cell adhesion properties of the hydrogel. To ensure data reliability, at least three independent images were measured for each experimental group, and the average cell adhesion area was calculated.

### 2.6. Evaluation of Angiogenic Potential

#### 2.6.1. Preparation of Extracts

The samples were initially rinsed with deionized water, then immersed in 75% ethanol for 24 h, followed by ultraviolet sterilization for 2 h. In accordance with ISO 10993 (Biological evaluation of medical devices—Part 12: Sample preparation and reference materials), the extract-volume-to-sample-surface-area ratio was maintained at 3:2, and the samples were submerged in α-MEM culture medium. The extract was incubated at 37 °C in a 5% CO_2_ environment. After 24 h, the extracts were obtained and filtered through a sterile filter, followed by storage at 4 °C.

#### 2.6.2. Tube Formation

HUVECs were seeded onto 6-well plates and incubated for 24 h before the extract was applied. The day prior to the experiment, Matrigel (Servicebio, Wuhan, China) was thawed on ice and allowed to melt slowly overnight at 4 °C. On the following day, 50 μL of Matrigel was added to each well of a 48-well plate, which was initially kept at 4 °C for 15 min, then equilibrated at room temperature for 30 min, and finally incubated at 37 °C for 45 min. When the HUVECs reached 70–80% confluence in the 6-well plate, they were digested, resuspended, and 50 μL of the cell suspension was added to each well of the 48-well plate. The plate was then incubated at 37 °C in a CO_2_ incubator. Tube formation was observed under a microscope at 2, 4, and 6 h. Finally, tube formation was assessed using an inverted microscope (IX51, Olympus, Tokyo, Japan), and branch points were quantitatively analyzed with ImageJ software. Triplicates of all experiments were performed.

#### 2.6.3. Cell Migration Experiment

HUVECs were seeded onto a 6-well plate and incubated for 24 h before adding the material extracts. Once the cells reached confluence, a 200 μL pipette tip was used to create a scratch in the center of each well, with a ruler placed vertically along the edge of the plate to ensure consistency. The wells were then washed three times with PBS to remove detached cells, and fresh medium was added. The plates were incubated at 37 °C with 5% CO_2_, and images were captured at 12 and 24 h. The images were subsequently analyzed using ImageJ software. Triplicates of all experiments were performed.

For the Transwell migration assay, HUVECs were prepared as a cell suspension at a concentration of 2 × 10^5^ cells/mL. A total of 700 μL of extract was added to the lower chamber, and 100 μL of the cell suspension was added to the upper chamber, followed by incubation at 37 °C for 24 h. After incubation, the Transwell inserts were washed three times with PBS, and the cells were fixed with 70% methanol for 30 min, then allowed to air dry. The cells were stained with 700 μL of 0.1% crystal violet solution for 15 min at room temperature, washed three times with PBS, and examined under an inverted microscope (IX71, Olympus, Tokyo, Japan). Cell migration was quantified by counting the cells in five randomly selected fields of view at ×100 magnification. Triplicates of all experiments were performed.

### 2.7. Evaluation of Osteogenic Differentiation

#### 2.7.1. Alkaline Phosphatase Activity Determination

BMSCs were seeded onto the samples and incubated for 7 and 14 days before undergoing ALP staining, which was performed for 15 min at room temperature. The purity of the total RNA extracted from the cells was assessed using spectrophotometry. DNA amplification was carried out using an ABI PRISM 7000 Sequence Detection System (Applied Biosystems, Foster City, CA, USA). The expression levels of osteogenesis-related genes, including bone sialoprotein (BSP), osteocalcin (OCN), osteopontin (OPN), and type I collagen (COL1A1), were measured. The GelMA hydrogel group was used as the control, with its expression normalized to 1-fold, and the experimental groups were compared to this baseline. All results were normalized to GAPDH expression.

#### 2.7.2. Osteogenic Differentiation

After sterilization, 2 × 10^6^ BMSCs were seeded onto the samples in 6-well plates. Once the cells reached 70–80% confluence, the medium was replaced with BMSC Osteogenic Induced Differentiation Medium (Procell, Wuhan, China), and the medium was refreshed every three days. After 7 days, the cells were washed three times with PBS and fixed with 4% paraformaldehyde for 15 min. Following fixation, the cells were stained with 1 mL of Alizarin red staining (Servicebio, Wuhan, China) for 15 min at room temperature. The cells were then washed three times with PBS, and the chamber was examined under an inverted microscope (IX71, Olympus, Tokyo, Japan). Cell counting was performed in five randomly selected fields of view at ×100 magnification. Triplicates of all experiments were performed.

### 2.8. Bacteriostatic Effect In Vitro

The *Staphylococcus aureus* and *E. coli* strains came from the Chinese Academy of Sciences Inspection and Quarantine Science Centre. LB medium was poured into the plate by pre-adding the bacterial solution. The composite material with a diameter of 6 mm was put into a Petri dish for the inhibition zone experiment, and finally, the size of the inhibition zone was compared by taking pictures.

### 2.9. In Vivo Study

#### 2.9.1. Experimental Procedures

The in vivo experiments were approved by the Animal Committee of the Tongren Hospital, Wuhan University (approval number: SY2022-021). There were 20 8-week-old male Sprague–Dawley (SD) rats with an average body weight of 180 g. The feeding conditions were 23 ± 2 °C, 50 ± 5% humidity, and 12 h light–dark cycle. All experimental procedures were carried out in accordance with the protocol approved by the Konkuk University Institutional Animal Care and Use Committee. A total of 20 Sprague-Dawley rats were randomly assigned to four groups, the control group, GelMA group, GelMA-C group, and GelMA-CM group, with five rats in each group. Anesthesia was induced by intraperitoneal injection of a ketamine (50 mg/kg) and xylazine (10 mg/kg) mixture. Once anesthesia was confirmed, the skulls were shaved and disinfected with povidone-iodine. A 3 cm incision was made along the sagittal suture to expose the bilateral parietal and occipital bones, separating the soft tissue and periosteum. Using a dental drill, two full-thickness bone defects, each 6 mm in diameter, were created bilaterally in the parietal bones. Continuous saline irrigation was applied throughout the procedure to prevent damage to the dura mater and brain tissue. After bone tissue removal, the samples were placed into the defects, and the periosteum and skin were sutured with resorbable stitches. The rats were then returned to their cages with free access to food and water. Following an 8-week implantation period, the rats were euthanized, and the parietal bones were harvested for subsequent analysis.

#### 2.9.2. Micro-Computed Tomography (Micro-CT) Analysis

After extraction and fixation of the cranial defects in 4% paraformaldehyde, all specimens were scanned using a SkyScan micro-CT instrument (Bruker, Billerica, MA, USA) at a resolution of 19.16 μm. The acquired image data were subsequently transferred to image analysis software. A cylindrical region of interest (ROI) was defined to encompass the original bone defect, and the new bone volume, bone volume/total volume (BV/TV), and bone mineral density (BMD) ratio within the ROI were calculated.

#### 2.9.3. Histological Analysis

The cranial defect samples were initially fixed in 4% paraformaldehyde, followed by decalcification with 10% EDTA and embedding in paraffin. Tissue sections, 10 µm in thickness, were then cut using a microtome and stained with Masson’s trichrome and hematoxylin and eosin (H&E). Section images were captured using an Olympus XM10 digital camera mounted on an Olympus BX51 microscope.

### 2.10. Statistical Analysis

Statistical analyses were conducted using a two-way analysis of variance (ANOVA), followed by Tukey’s post hoc test for multiple comparisons. We conducted the Shapiro–Wilk test for normality on the data for all groups prior to performing the two-way ANOVA. This test confirmed that the data followed a normal distribution, allowing us to proceed with the parametric statistical tests. Data are presented as the mean ± standard deviation, and comparisons were made across all experimental groups.

## 3. Results and Discussion

### 3.1. Preparation and Characterization of Hydrogel

At room temperature, a photoinitiator, MPC powder, and CMCS powder were added into the GelMA. The pre-gel mixture was heated in a water bath at 60–70 °C until the GelMA dissolved completely. The mixture can remain liquid in a water bath at 37 °C. As shown in Figure 2A, the samples underwent three processes before testing: pre-cooling, liquid nitrogen fracture, and lyophilization. The composition of the composite was confirmed by FT-IR (Figure 2C) and XRD (Figure 2B) analysis. The presence of C-O-C stretching and differences in the C-H stretching and bending regions, compared to gelatin, suggest the existence of the methacrylate group in the GelMA monomer [22]. Additionally, the shift of the amide I peak to 1076 cm⁻^1^ provides evidence for the formation of amide linkages. After the addition of carboxymethyl cellulose, the hydroxyl O-H stretching vibration signal increased at 3278 cm⁻^1^, the antisymmetry and symmetry of COO^⁻^ at 1637 and 1402 cm⁻^1^, and the cellulose C-O at 1028 cm⁻^1^. After the addition of MPC, the antisymmetric telescopic vibration signal of phosphate PO_4_^3^⁻ was increased to 1028 cm⁻^1^. The diffraction peak positions in the sample are essentially consistent with those in the standard PDF card of MPC, confirming the presence of hexahydrated magnesium phosphate (KMgPO_4_·6H_2_O).

An ideal bone graft material (BGM) must preserve its shape and mechanical properties, as it replaces damaged tissue and facilitates bone regeneration. The viscoelastic modulus of the GelMA, GelMA-C, and GelMA-CM samples was measured within a frequency range of 0.1–10 Hz (Figure 3D). The storage modulus of GelMA, GelMA-C, and GelMA-CM was measured as 10.349 ± 0.108, 10.874 ± 0.064, and 14.428 ± 0.440 kPa, respectively. Additionally, the loss modulus was measured as 0.055 ± 0.025, 0.207 ± 0.123, and 0.339 ± 0.077 kPa, respectively. The loss tangent graph of all samples increases with frequency (Figure 2E). In previous studies conducted by our research group, the viscoelastic modulus of GelMA hydrogel composites with 7.5%, 5%, and 2.5% MPC was 2.1483 ± 0.0164, 1.3880 ± 0.0318, and 0.8537 ± 0.0878 kPa, respectively. The addition of CMCS improved the mechanical strength of the hydrogel. A greater storage modulus indicates a material that is more elastic in nature, while a higher loss modulus signifies a viscous, fluid-like behavior with less elasticity. Furthermore, an increase in the loss tangent signifies a decrease in elasticity [20,23]. The storage modulus (G’), which reflects the elasticity of a material, is an important parameter for tissue engineering, particularly for bone tissue regeneration. A higher G’ indicates better structural integrity, which is critical for scaffolds that need to support tissue growth and integration. The improvement observed in the GelMA-CM composite (14.428 ± 0.440 kPa) compared to GelMA and GelMA-C suggests that the addition of CMCS enhances the hydrogel’s stiffness, making it more suitable for applications where mechanical support is required, such as in bone defect repair. The loss modulus (G″), which reflects the viscoelastic behavior of a material, indicates the damping capacity of a hydrogel. The increase in G″ values in GelMA-CM suggests that the hydrogel could also exhibit better deformability and shock-absorbing properties, which are important for materials designed to withstand mechanical forces in dynamic physiological environments, such as bone. Unlike other BGMs, GelMA-CM can resist some external compression when implanted in a bone defect. While the current study addresses the mechanical properties of hydrogels suitable for non-load-bearing applications, our future efforts will aim to further optimize the material’s mechanical strength and biocompatibility to create scaffolds capable of withstanding the mechanical stresses found in weight-bearing bone regions. This includes exploring composite materials and reinforcement strategies to enhance stiffness and structural integrity while maintaining the bioactive properties necessary for tissue regeneration.

The samples were crosslinked and solidified after UV irradiation. The morphology of the samples was characterized using FE-SEM (Figure 2F). MPC was observed as particles ranging from 20 to 70 µm in size, while the GelMA hydrogel exhibited a sponge-like structure with pores between 5 and 100 µm in diameter. The porous structure of the hydrogel was formed during freeze drying through the sublimation of water.

### 3.2. In Vitro Cytocompatibility Evaluation of Hydrogel

Evaluating cytocompatibility is essential for scaffolds in bone regeneration, with methods varying based on the properties of the materials and coculture systems [24,25,26]. In vitro, cells were cultured in liquid media, which required the scaffold’s dispersing ability. Most commonly, cells are seeded directly on the scaffold surface or cultured with a leaching solution. As of yet, a comprehensive assessment of cell adhesion has not been completed. As shown in Figure 3A, we prepared extracts to evaluate cytocompatibility. rBMSCs were cultured on extracts for 1, 3, and 7 days, followed by analysis using the EZ-Cytox assay (Figure 3B). Viability values were compared to the 1-day GelMA group. The results indicated that the viability values for the GelMA-CM group were the highest, at 131.4%, 186.8%, and 455.8% on days 1, 3, and 7, respectively. Additionally, the cell viability rates for the GelMA-C group were 160.8% and 369.2% on days 3 and 7, respectively. Chitosan (CS), a partially deacetylated derivative of chitin, has a structure similar to glycosaminoglycans (GAGs), an essential component of the extracellular matrix (ECM). Carboxymethyl chitosan (CMC) is one of the most common chitosan derivatives, known to enhance biocompatibility. When HUVECs were cultured with the extracts, the conditions of live and dead cells and the arrangement of the cell cytoskeleton were examined, as shown in Figure 3C. In general, cells cocultured with GelMA-CM exhibited abundant branched long filopodia, indicating better biocompatibility.

### 3.3. In Vitro Osteogenic Property Evaluation of Hydrogel

For over 60 years, alkaline phosphatase (ALP), an ectoenzyme found on the surface of osteoblasts, has been widely recognized as a fundamental marker for bone metabolism assessment [27]. The ALP activities of the samples after 14 days are shown in Figure 4A. ALP activity at 7 days did not differ significantly between the GelMA and GelMA-C groups, but an enhancement was observed in the GelMA-CM group. At 14 days, the ALP activity in the GelMA-CM group was higher compared to the other group (*p* < 0.001) (Figure 4C). The heightened ALP activity reflects the significant hydrolysis of organic phosphatase, leading to the release of free inorganic phosphate and further advancing matrix mineralization compared to the other samples. Alizarin red staining, which reacts with calcium to produce dark red compounds, was used to visualize osteogenesis-induced calcium nodules. The addition of MPC at 14 days led to an increase in mineral deposition, as clearly demonstrated by the Alizarin red staining (Figure 4B), consistent with the quantitative findings from the ARS analysis (Figure 4D). The gradual increase in red calcium nodules indicates that the GelMA-CM group possesses osteogenic potential. The osteogenic differentiation effect of GelMA-CM was further confirmed by RT-PCR analysis. GelMA-CM significantly promoted the expression of osteogenesis-related genes, including COL1, OPN, OCN, and BSP, which are crucial for bone formation [28,29]. COL1 and OPN are primarily expressed during early bone formation, whereas OCN and BSP are more prominent during the late stages of bone maturation [20,23]. This gene expression timing was reflected in our experimental results. After 7 days of rBMSC differentiation in the GelMA-CM group, COL1 and OPN expression levels were 7.5 and 3.45 times higher than those in the control group, respectively (Figure 4E,F), while OCN and BSP levels were higher on the 14th day (Figure 4G,H). By introducing MPC, Mg^2^⁺ ions were released, which subsequently promoted bone formation once a specific concentration was achieved [23]. Mg^2^⁺ is thought to promote the proliferation and osteogenic differentiation of MSCs by sequentially activating the MAPK/ERK and Wnt/β-catenin signaling pathways [30]. In our study, all osteoblast-related genes exhibited the highest expression levels in the GelMA-CM group, while the gene expression levels in the GelMA and GelMA-C groups were similar to or slightly higher than those in the control group.

### 3.4. In Vitro Angiogenic Property Evaluation of Hydrogel

Angiogenesis, the formation of new blood vessels from existing capillaries or postcapillary veins, plays a crucial role in various biological processes, including organ growth, embryonic development, and wound healing. Wound healing assays are commonly employed to observe the effects of exogenous factors, such as drugs and genes, on cell migration, repair, and interaction. In this study, the GelMA-CM group exhibited a significantly enhanced cell migration capacity compared to other groups (Figure 5A,B), with a statistically significant difference (*p* < 0.05) (Figure 5E). Transwell assays are often utilized to study cell migration, chemotaxis, and invasion in response to various stimuli, including growth factors, chemokines, and extracellular matrix components. The results demonstrated that the invasive capacity of cells in the GelMA and GelMA-C groups was comparable, with both groups showing slightly higher invasion levels than the control group (Figure 5C). However, upon the addition of CMCS, the invasive capacity of the cells increased significantly (Figure 5F). The endothelial cells are capable of dividing and migrating rapidly when angiogenic signals are present. Currently, blood vessel formation is often studied through lumen formation assays, where endothelial cells form cords that eventually develop into lumens, a process that can be replicated under specific in vitro conditions (e.g., using Matrigel or collagen). Luminal formation is an early indicator of capillary development and reflects the functional integrity of endothelial cells in vitro. In this experiment, cells were dispersed at 2 h, with no vascular-like structures observed. By 6 h, the cells formed branches and interconnected to create a complete reticular structure. Over time, the cells began to cluster, causing the network-like structure to dissipate. Therefore, 6 h was selected as the optimal observation time point for this experiment (Figure 5D). Branch points were quantified to assess the angiogenic capacity of the samples. The GelMA-CM group exhibited a significantly higher number of branch points compared to the control group (Figure 5G), indicating its angiogenic potential.

The enhancement in cell invasion and angiogenesis by carboxymethyl chitosan (CMCS) may involve several key mechanisms. Firstly, CMCS, as a derivative of chitosan, possesses carboxyl groups that can interact with extracellular matrix (ECM) components, such as collagen and glycosaminoglycan. This interaction may alter the ECM structure, promoting cell adhesion, migration, and proliferation. Secondly, CMCS may act as a chemotactic agent, activating specific cell surface receptors, such as integrins, which enhance cell attachment to the ECM and promote migration. Furthermore, CMCS has the potential to stimulate endothelial cell migration and tube formation, key steps in angiogenesis, possibly by upregulating angiogenic growth factors such as VEGF and bFGF. These mechanisms suggest that CMCS plays a multifaceted role in enhancing cell invasion and angiogenesis, warranting further investigation to elucidate the exact pathways involved. However, as this is a complex and multifactorial process, we acknowledge that further investigation is required to comprehensively understand the exact mechanisms involved.

### 3.5. Antimicrobial Performance of Hydrogel

There is a significant public health concern associated with the indiscriminate use of antibiotics [31]. Consequently, antibiotic usage needs to be controlled [32]. The use of antibacterial biomaterials is one way to reduce antibiotic dependence. Antibacterial tissue engineering scaffolds and other implantable biomaterials have been the focus of recent research [33,34]. There are several advantages to incorporating antibacterial properties into biomaterials during cell therapy, including reducing the risk of infection. Due to its positively charged protonated amino groups within a polymer chain, chitosan has gained widespread use in antibacterial materials [35]. These amino cations interfere with the synthesis of macromolecular substances on bacterial cell membranes, altering membrane permeability.

The inhibition zone method, which assesses the antibacterial potency of drugs, involves the diffusion of a drug on an agar plate, inhibiting bacterial growth and forming a transparent circle around the area. This method was employed in the experiment using *Escherichia coli* and *Staphylococcus aureus*, each with a bacterial concentration of 10^6^ cells/mL. As shown in the figure, a transparent circle was observed in all three groups (Figure 6A), with the addition of CMCS enhancing the antibacterial capability of the composites (Figure 6B).

CMCS has been widely reported for its antibacterial properties, which are thought to be mediated by several mechanisms. Firstly, the positively charged amino groups on the CMCS molecule can interact with the negatively charged bacterial cell membranes, leading to membrane disruption and subsequent bacterial cell death. Secondly, CMCS may induce oxidative stress in bacterial cells, promoting the generation of reactive oxygen species (ROS), which can damage cellular structures such as proteins, lipids, and DNA. Additionally, CMCS may interfere with bacterial cell wall synthesis or inhibit essential enzymatic activities, further contributing to its antibacterial effects.

Although these mechanisms are suggested by previous studies, the exact pathways through which CMCS exerts its antibacterial activity in our specific system remain to be fully elucidated. We acknowledge that more detailed research is needed to better understand these mechanisms in the context of our study. Therefore, we plan to investigate these mechanisms in future research, including examining bacterial adhesion, membrane integrity, and the generation of ROS in the presence of CMCS.

### 3.6. In Vivo Osteogenic Property Evaluation of Hydrogel

The composites were injected into the defect site of the rat skull and subsequently crosslinked under UV light. Zhang et al. reported that Mg^2+^ promotes osteoblast recruitment and osteogenesis by activating PI3K phosphorylation via TRPM7 channels [36]. However, other divalent cations, particularly Ca^2^⁺, also play a crucial role in bone regeneration [37]. The GelMA-CM retained the reconstructed shape at the bone defect site without requiring additional equipment and controlled the absorption rate. GelMA hydrogel was still present up to 8 weeks after implantation, whereas the other two groups were almost completely degraded by this time. The degradation of composites occurs through two primary mechanisms: (1) a rapid degradation of polymeric backbones or cleavage of hydrogel crosslinking bonds and (2) a slower process involving the hydrolysis and dissolution of the hydration product for MPC [38]. The different degradation rates of the composites may be closely related to the rapid hydrolysis and high swelling ratio of the GelMA-C hydrogel. GelMA-C hydrogel exhibited exceptional osteoconductivity, promoting bone regeneration which integrated seamlessly with the original bone and significantly enhanced regeneration by increasing the expression of osteoblastic markers.

The osteogenic properties of the control, GelMA, GelMA-C, and GelMA-CM groups were evaluated in vivo using critically sized calvarial defect models in Sprague-Dawley rats. Specifically, samples were sequentially implanted into the corresponding positions and allowed to reconstruct over 8 weeks (Figure 6C). As shown in Figure 6D, micro-CT imaging revealed significantly enhanced new bone growth in the GelMA-CM group compared to the control group at 8 weeks. Quantitative analysis further confirmed this trend, with the GelMA-CM group showing 1.33-fold higher BV/TV and 1.65-fold greater new bone volume compared to the GelMA-C group.

The defects were sectioned perpendicularly along the central line of the defect area for histological examination. Masson’s trichrome staining and H&E staining were used to identify fibrous tissue, inflammation, newly formed bone, and original bone. In the defect-only group, the defected area was predominantly filled with fibrous connective tissue, and only a small amount of bone formation occurred near the edges. In the GelMA group, remnants of the GelMA hydrogel were clearly visible, and newly formed bone was integrated with the original edges of the bone defect. In contrast, the GelMA-C group showed small osseous islands, with remnants of GelMA-C hydrogel becoming calcified and surrounded by newly formed ossifying bone after 8 weeks. The amount of new bone tissue in this group was increased compared to the GelMA group. In contrast, in the GelMA-CM group, regenerated bone covered most of the defect area (Figure 7). Studies have demonstrated that the concentration of Mg^2^⁺ must be carefully adjusted within an optimal range (below 15 mmol/L) to effectively promote osteogenic activity; higher levels of Mg^2^⁺ may not promote osteogenic differentiation of osteoblasts [39]. GelMA-CM releases a lower content of Mg^2^⁺ compared to existing Mg-based materials, such as Mg-ion screws and ceramic alloy materials, resulting in less inhibition of bone mineral formation. This approach may allow for the post-curing repair of small-site defects in non-load-bearing bones through injection, simplifying the procedure and reducing clinical costs without the need for surgery.

After a successful bone implant, three stages of new bone formation occur as the following [40]: the process progresses from an initial stage of coagulation and acute inflammation, followed by a transient phase of bone formation that involves chronic inflammation and osteogenesis, ultimately leading to bone remodeling. Our experimental results suggest that GelMA-CM has the potential to be used as synthetic bone graft material (BGM), though further improvement is needed to address potential inflammatory responses. Future studies should explore targeted administration and controlled release of Mg^2^⁺, the relationship between GelMA-CM and inflammatory response, and the effect of GelMA-CM on the pattern of new bone growth.

## 4. Conclusions

We developed a GelMA-CM composite for bone regeneration. This composite can be injected into bone defects, solidify under UV light, and retain its shape at the defect site without requiring additional equipment. The incorporation of CMCS into the GelMA hydrogel enhanced the material’s mechanical strength and viscoelastic properties, making it more suitable for applications in bone tissue engineering. The hydrogel composite demonstrated good cell viability and cell adhesion, supporting its potential for use as a scaffold in bone regeneration. In this study, we focused primarily on the osteogenic potential of the GelMA-CM composite, and while preliminary results suggest that CMCS may contribute to enhanced angiogenesis and antibacterial effects, we acknowledge that these mechanisms require more in-depth exploration. As part of our future research, we plan to investigate the following: the angiogenic mechanism of CMCS, including its effect on endothelial cell behavior and vascularization within the scaffold, which will be crucial for improving the blood supply to the regenerating tissue, and the antibacterial properties of CMCS, exploring its potential to inhibit bacterial growth and prevent infection, which is especially important for materials used in bone regeneration and surgical applications. Our future efforts will also aim to further optimize the material’s mechanical strength and biocompatibility to create scaffolds capable of withstanding the mechanical stresses found in weight-bearing bone regions. This includes exploring composite materials and reinforcement strategies to enhance stiffness and structural integrity while maintaining the bioactive properties necessary for tissue regeneration.

## Figures and Tables

**Figure 1 biomedicines-13-00257-f001:**
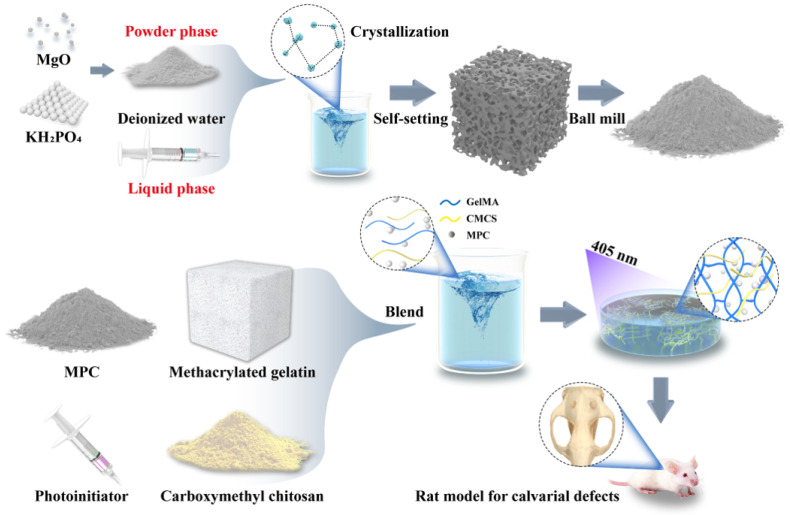
Schematic illustration of the preparation process of magnesium phosphate cement (MPC), assembly of GelMA-CM composite material, and its application in bone defect repair.

**Figure 2 biomedicines-13-00257-f002:**
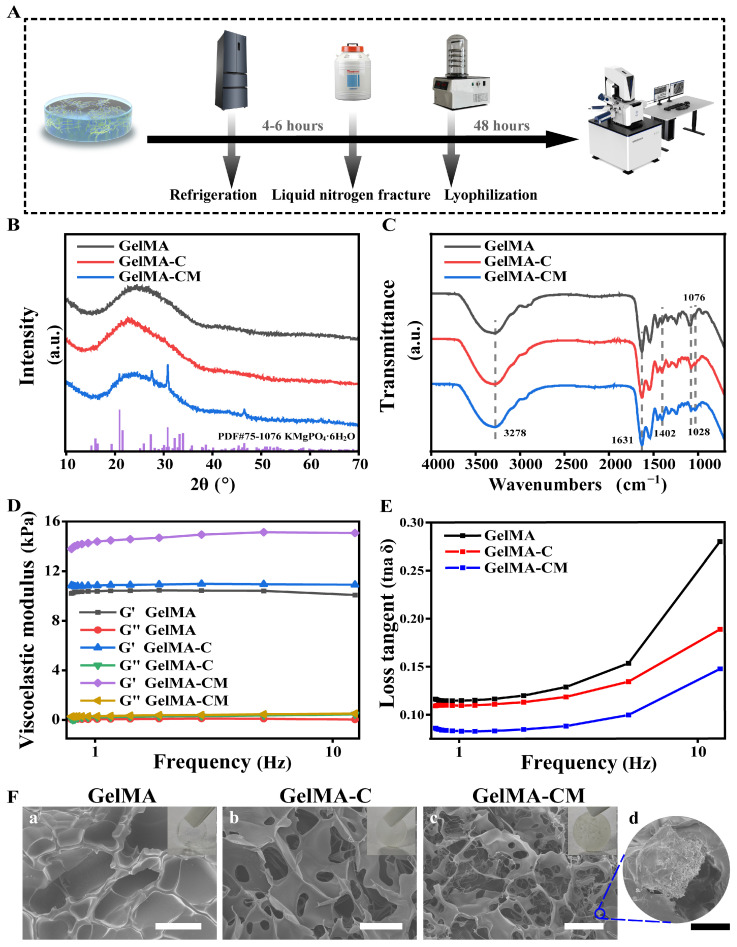
The effects of GelMA, GelMA-C, and GelMA-CM on physical and chemical characterization. (**A**) Schematic image of hydrogel pre-treatment. (**B**) XRD and (**C**) FT-IR spectra of samples. (**D**) Dependence of storage modulus (G′) and loss modulus (G″) on the frequency of samples. (**E**) The viscoelastic modulus of samples was calculated and compiled as loss tangent values. (**F**) a–c show digital and FE-SEM images of three samples, and d shows a magnified view of GelMA-CM. The scale bars for white and black are 100 µm and 20 µm, respectively.

**Figure 3 biomedicines-13-00257-f003:**
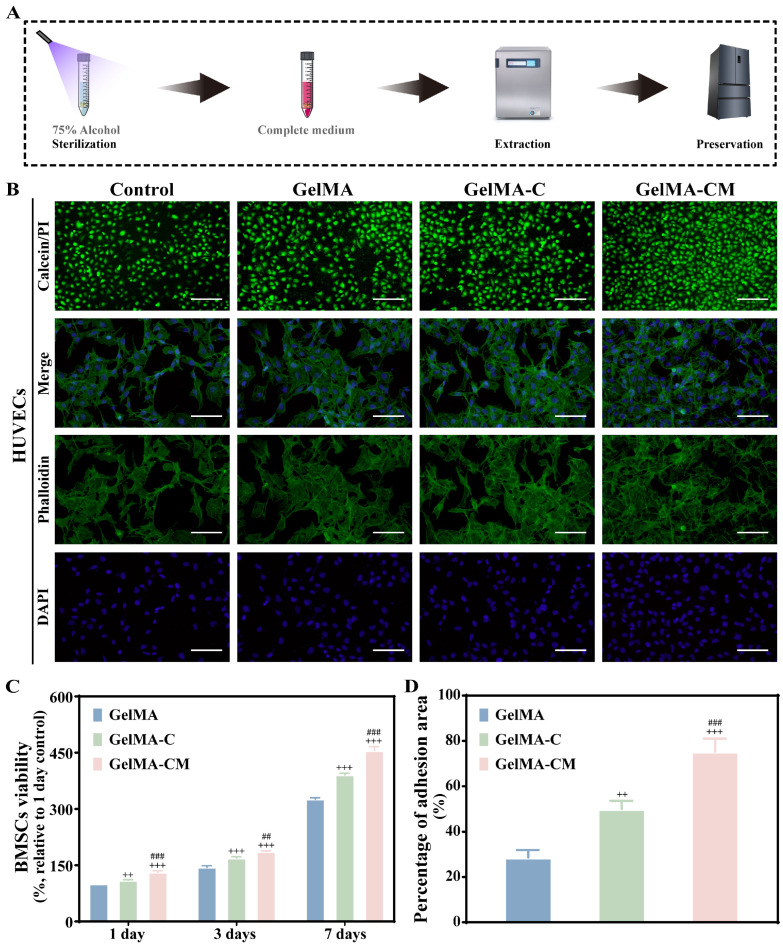
Effects of GelMA, GelMA-C, and GelMA-CM on cytocompatibility in vitro. (**A**) Schematic diagram of extract preparation. (**B**) The effect of samples on HUVEC adhesion and proliferation. In live/dead staining, green staining was used for live cells, while red staining was used for dead cells. Phalloidin staining and DAPI staining were used to visualize living and stationary cells in blue. (**C**) In vitro viability evaluation of rBMSCs in GelMA hydrogel at 1, 3, and 7 days. (**D**) Quantitative analysis of adhesion area. Scale bars are 100 µm. Results are expressed as the mean ± SD of triplicate experiments: ++ *p* < 0.01, and +++ *p* < 0.001 indicate significant differences compared to the GelMA group. ## *p* < 0.01, and ### *p* < 0.001 indicate significant differences compared to the GelMA-C group.

**Figure 4 biomedicines-13-00257-f004:**
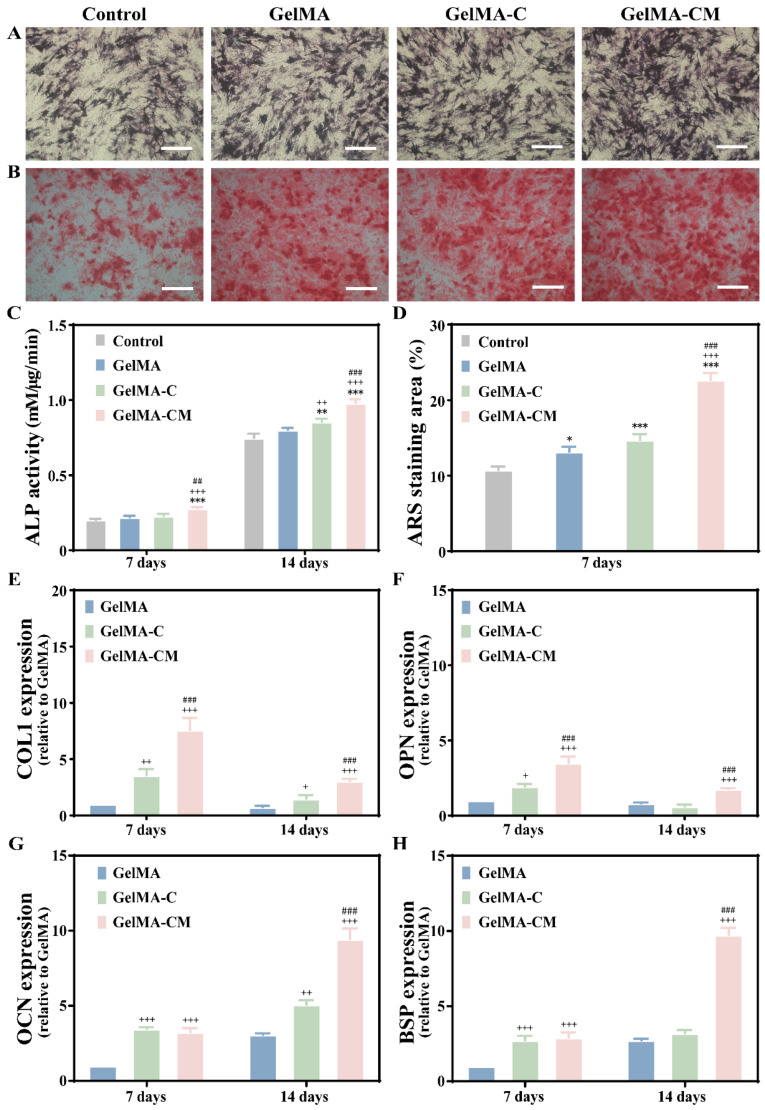
Effects of GelMA, GelMA-C, and GelMA-CM on osteogenic differentiation in vitro. (**A**) ALP staining and (**C**) quantitative analysis of bone marrow mesenchymal stem cells co-cultured for 14 days. (**B**) Alizarin red staining and (**D**) quantitative analysis of cells cultured for 7 days. (**E**–**H**) Expression of bone-related genes on the GelMA, GelMA-C, and GelMA-CM hydrogels for 7 days and 14 days, respectively. Scale bars are 100 µm. Results are the mean ± SD of triplicate experiments: + *p* < 0.05, ++ *p* < 0.01, and +++ *p* < 0.001 indicate significant differences compared to the GelMA group. ## *p* < 0.01, and ### *p* < 0.001 indicate significant differences compared to the GelMA-C group. * *p* < 0.05, ** *p* < 0.01, and *** *p* < 0.001 indicate significant differences as compared with the control group.

**Figure 5 biomedicines-13-00257-f005:**
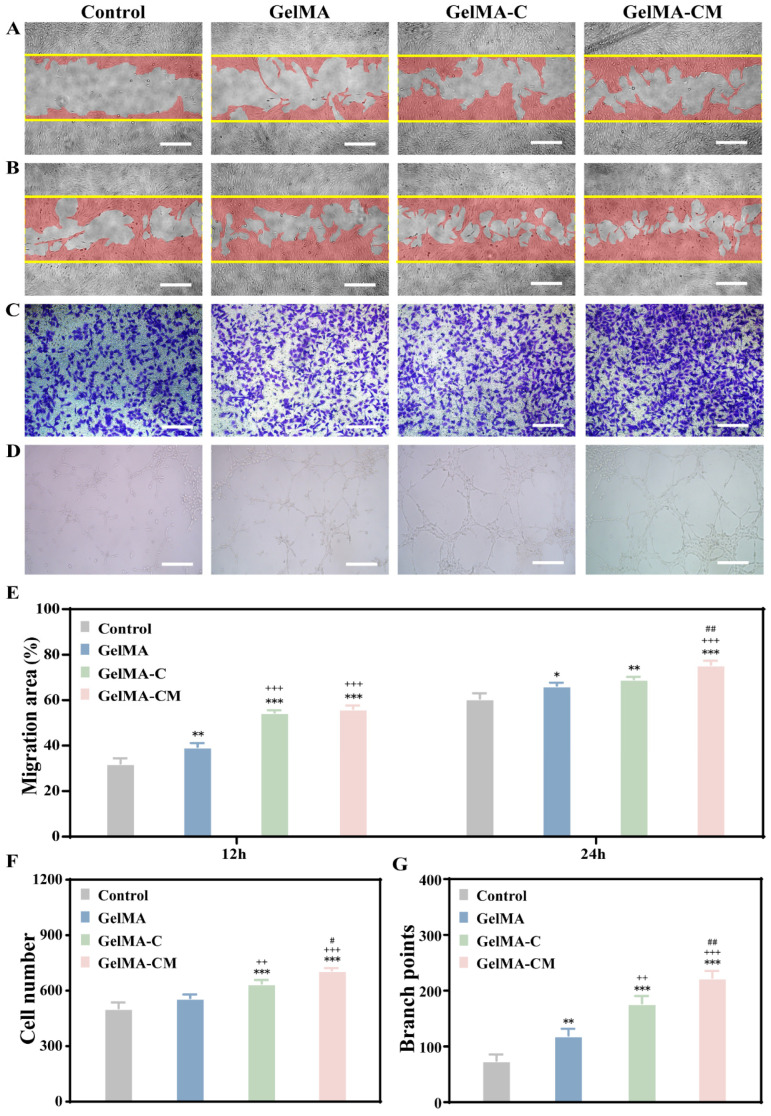
Effects of GelMA, GelMA-C, and GelMA-CM on angiogenesis in vitro. (**A**,**B**) Results of 12 and 24 h scratch experimental and (**E**) quantitative analysis. (**C**) Transwell results and (**F**) quantitative analysis of bone marrow mesenchymal stem cells co-cultured with composite extract. (**D**) Tube formation assay after 8 h of co-culture of MSCs and material extract. (**G**) The number of branch points in tube formation. Scale bars are 100 µm. Results are the mean ± SD of triplicate experiments: ++ *p* < 0.01, and +++ *p* < 0.001 indicate significant differences compared to the GelMA group. # *p* < 0.05, and ## *p* < 0.01 indicate significant differences compared to the GelMA-C group. * *p* < 0.05, ** *p* < 0.01, and *** *p* < 0.001 indicate significant differences as compared with the control group.

**Figure 6 biomedicines-13-00257-f006:**
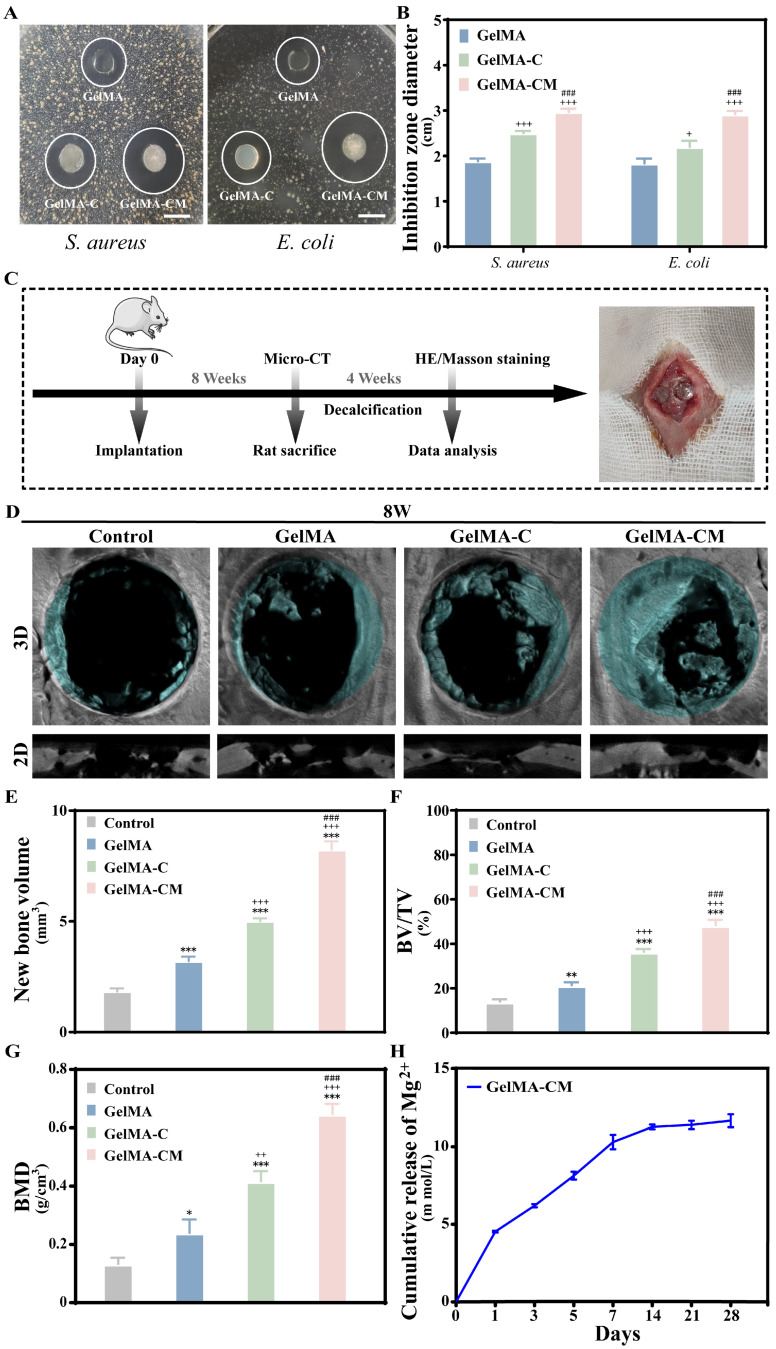
Effects of GelMA, GelMA-C, and GelMA-CM on antibacterial properties and osteogenesis in vivo post-surgery for 8 weeks. (**A**) Digital images and (**B**) quantitative analysis of inhibition zones for samples. (**C**) An illustration of the experimental design and the implantation process used to evaluate in vivo osteogenesis. (**D**) Representative 3D reconstruction micro-CT images. (**E**–**G**) Analysis of microstructural parameters such as new bone volume, BV/TV, and BMD, quantitatively. (**H**) Ion release of magnesium ions in the medium within 4 weeks. Results are the mean ± SD of triplicate experiments: + *p* < 0.05, ++ *p* < 0.01, and +++ *p* < 0.001 indicate significant differences compared to the GelMA group. ### *p* < 0.001 indicate significant differences compared to the GelMA-C group. * *p* < 0.05, ** *p* < 0.01, and *** *p* < 0.001 indicate significant differences as compared with the control group. Scale bar: (**A**) 0.4 cm for inhibition zones, (**D**) 50 μm for 3D images.

**Figure 7 biomedicines-13-00257-f007:**
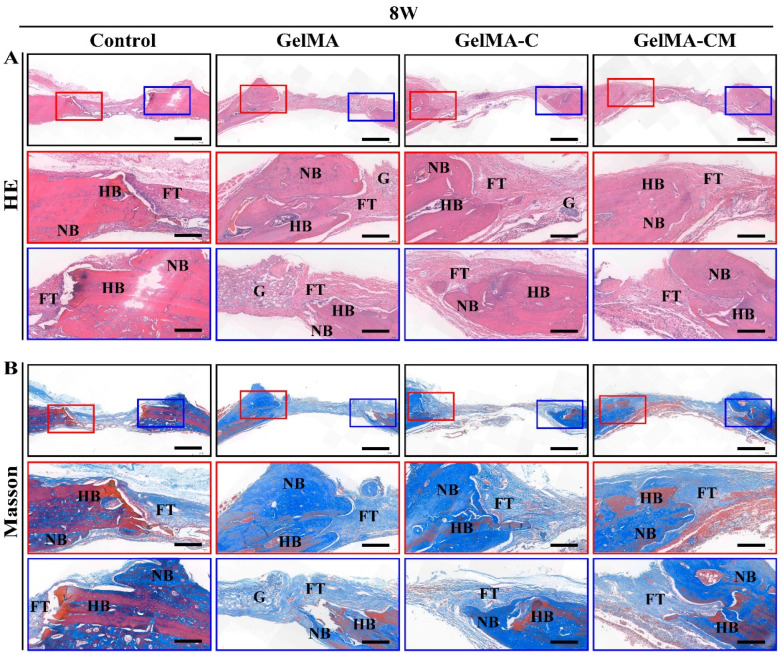
Effects of samples on osteogenesis in vivo post-surgery for 8 weeks. (**A**) Representative H&E staining images. (**B**) Masson staining images. The red and blue boxes highlighting and magnifying two different regions, respectively. Scale bar: 500 μm for low magnification and 100 μm for high magnification. (G: GelMA, FT: Fibrous tissues, HB: Host bone tissues, NB: New bone tissues).

## Data Availability

The original contributions presented in this study are included in the article. Further inquiries can be directed to the corresponding author.

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
