# Peer review of "Functionalized GelMA/CMCS Composite Hydrogel Incorporating Magnesium Phosphate Cement for Bone Regeneration"

_biomedicines, 2025, doi:10.3390/biomedicines13020257_

Round 1

Reviewer 1 Report

Comments and Suggestions for Authors
The manuscript entitled "Functionalized GelMA/CMCS Composite Hydrogel Incorporating Magnesium Phosphate Cement for Bone Regeneration" is a well written and conducted study. In order to further consider for publication in this journal, the following comments must be addressed by the authors

1- The reported cell viability values exceeding 100% (e.g., 131.4%, 186.8%, 455.8%) are highly unusual and raise serious concerns. Cell viability assays typically report values between 0% and 100%. Values above 100% suggest a potential error in the assay, data analysis, or a lack of appropriate controls (e.g., a negative control without any extract). The text needs to address this discrepancy and explain how these values were obtained and validated. Without a proper explanation, these results are not credible.

2- The description of cell morphology ("abundant branched long filopodia, indicating better biocompatibility") is subjective and lacks quantitative support. While filopodia can indicate cell adhesion and spreading, their abundance and branching alone are not definitive proof of "better biocompatibility." The abstract needs to provide more objective measures of biocompatibility, such as quantification of cell adhesion, spreading area, or expression of specific cell adhesion molecules. Relying solely on visual interpretation of cell morphology is insufficient for a robust conclusion about biocompatibility. Furthermore, the comparison group for this morphological assessment is unclear.

3- The author mentioned that the addition of MPC (presumably a magnesium-containing compound) leads to Mg²⁺ release and subsequent bone formation. However, critical details are missing. What is the specific composition of MPC? What is the concentration of MPC used? How was the release of Mg²⁺ ions measured and quantified?
Without this information, the conclusions about the role of MPC in enhancing osteogenesis remain speculative. The statement that Mg²⁺ "promoted bone formation once a specific concentration was achieved" is vague and needs quantitative support demonstrating this concentration dependent effect.

4- Authors repeatedly uses strong language ("superior osteogenic capacity," "significantly higher," "significantly promoted") based on what appears to be a limited set of presented data (Fig. 4A-H). While the results suggest a positive effect of GelMA-CM, the absence of detailed statistical analysis (p-values, error bars, sample sizes) in the description weakens the strength of the conclusions. I cannot assess the statistical significance of the observed differences. Furthermore, the text only mentions the GelMA-CM group showing superior results compared to a control, without providing a full comparison across all groups (GelMA, GelMA-C, GelMA-CM) for all measured parameters at both time points. This lack of comprehensive data presentation limits the ability to draw robust conclusions about the relative effects of each experimental condition

5- Authors states that adding CMCS (presumably a chitosan-based material) significantly increased invasive capacity (Fig. 5F) and angiogenic potential (Fig. 5G). However, it doesn't explain how CMCS achieves this. Is it providing a chemotactic signal? Does it alter the extracellular matrix in a way that promotes migration and tube formation?
The mechanism by which CMCS enhances cell invasion and angiogenesis is entirely absent, leaving the reader to speculate on the underlying biological processes. Simply stating a statistical significance without elucidating the mechanism weakens the scientific contribution.

6- Authors states that chitosan's positively charged amino groups interfere with bacterial cell membrane synthesis, altering membrane permeability. While this is a common explanation, it's an oversimplification. The exact mechanism of chitosan's antibacterial action is complex and multifaceted, involving interactions with multiple bacterial components and pathways. It's likely not solely about membrane permeability alteration. A more nuanced description acknowledging the complexity of the mechanism would improve the scientific accuracy of the text. Mentioning other potential mechanisms, such as cell wall disruption or interaction with bacterial DNA, would be beneficial.

Author Response

1- The reported cell viability values exceeding 100% (e.g., 131.4%, 186.8%, 455.8%) are highly unusual and raise serious concerns. Cell viability assays typically report values between 0% and 100%. Values above 100% suggest a potential error in the assay, data analysis, or a lack of appropriate controls (e.g., a negative control without any extract). The text needs to address this discrepancy and explain how these values were obtained and validated. Without a proper explanation, these results are not credible.
Reply: Thank you for your valuable feedback.

We would like to clarify the reason behind the cell viability values exceeding 1 in our study.

In our experimental setup, cell viability was measured using EZ-Cytox, and the values presented in the manuscript are relative to the control group. The control group was used as a baseline, with its cell viability set at 100% (or a value of 1).

As a result, cell viability values greater than 1 indicate an increase in cell viability relative to the control, meaning the experimental treatment led to a higher level of cell survival or proliferation compared to the untreated control. For instance, a value of 1.2 corresponds to a 20% increase in cell viability compared to the control group.

This approach allows us to assess the effects of different experimental conditions in a standardized way, providing a clearer comparison between the groups. We hope this explanation clarifies the basis for the values greater than 1 in our results.

Thank you again for your insightful feedback. We look forward to your further comments and suggestions.

2- The description of cell morphology ("abundant branched long filopodia, indicating better biocompatibility") is subjective and lacks quantitative support. While filopodia can indicate cell adhesion and spreading, their abundance and branching alone are not definitive proof of "better biocompatibility." The abstract needs to provide more objective measures of biocompatibility, such as quantification of cell adhesion, spreading area, or expression of specific cell adhesion molecules. Relying solely on visual interpretation of cell morphology is insufficient for a robust conclusion about biocompatibility. Furthermore, the comparison group for this morphological assessment is unclear.
Reply: Thank you for your valuable feedback.

We appreciate your recommendation to include quantitative data on cell adhesion area, and we have now addressed this point in our revised manuscript.

In response to your suggestion, we have performed additional experiments to quantify the cell adhesion area. The results have been added to the revised manuscript in the Results section "Section 2.5.2, Figure 3D", where we present a detailed comparison of cell adhesion between the experimental and control groups.

Thank you again for your insightful feedback. We look forward to your further comments and suggestions.

3- The author mentioned that the addition of MPC (presumably a magnesium-containing compound) leads to Mg²⁺ release and subsequent bone formation. However, critical details are missing. What is the specific composition of MPC? What is the concentration of MPC used? How was the release of Mg²⁺ ions measured and quantified?

Without this information, the conclusions about the role of MPC in enhancing osteogenesis remain speculative. The statement that Mg²⁺ "promoted bone formation once a specific concentration was achieved" is vague and needs quantitative support demonstrating this concentration dependent effect.

Reply: Thank you for your valuable feedback.

The MPC used in our study is composed of hexahydrated magnesium potassium phosphate (KMgPO4·6H2O). We selected a concentration of 5% based on our group’s prior experiments, where this concentration has shown optimal performance in similar applications related to bone regeneration.

Additionally, as per your suggestion, we have included an experiment to measure the magnesium ion (Mg²⁺) release from the MPC. In our previous manuscript submission, this data was not presented because only a single experimental group was used. However, we have now added the relevant magnesium ion release data in the revised manuscript "Section 2.4, Figure 6H". This data provides a clearer understanding of the material’s behavior and its potential impact on the biological environment, especially in terms of supporting bone regeneration.

Thank you again for your insightful feedback. We look forward to your further comments and suggestions.

4- Authors repeatedly uses strong language ("superior osteogenic capacity," "significantly higher," "significantly promoted") based on what appears to be a limited set of presented data (Fig. 4A-H). While the results suggest a positive effect of GelMA-CM, the absence of detailed statistical analysis (p-values, error bars, sample sizes) in the description weakens the strength of the conclusions. I cannot assess the statistical significance of the observed differences. Furthermore, the text only mentions the GelMA-CM group showing superior results compared to a control, without providing a full comparison across all groups (GelMA, GelMA-C, GelMA-CM) for all measured parameters at both time points. This lack of comprehensive data presentation limits the ability to draw robust conclusions about the relative effects of each experimental condition
Reply: Thank you for your valuable feedback.

The osteogenic performance in our study was assessed using several well-established techniques, including alkaline phosphatase (ALP) activity, Alizarin Red S (ARS) staining for mineralization, and the evaluation of osteogenesis-related proteins. These methods were chosen to provide a comprehensive understanding of the material’s ability to promote bone formation at both the early and later stages of osteogenic differentiation. We have carefully considered your feedback regarding the wording used in the manuscript and have made the necessary revisions. Specifically, we have refined the language to ensure clarity and precision in the description of osteogenic performance.

To ensure the robustness and statistical significance of our findings, we have included error bars in the relevant figures to represent the standard deviation of the data. Additionally, we performed inter-group comparisons to highlight any significant differences between experimental groups. These statistical analyses allow us to draw reliable conclusions regarding the osteogenic potential of the tested materials.

The results are presented in the revised manuscript in " Section 3.3, Figure 4", where you can find both the quantitative data and the corresponding statistical comparisons.

Thank you again for your insightful feedback. We look forward to your further comments and suggestions.

5- Authors states that adding CMCS (presumably a chitosan-based material) significantly increased invasive capacity (Fig. 5F) and angiogenic potential (Fig. 5G). However, it doesn't explain how CMCS achieves this. Is it providing a chemotactic signal? Does it alter the extracellular matrix in a way that promotes migration and tube formation?

The mechanism by which CMCS enhances cell invasion and angiogenesis is entirely absent, leaving the reader to speculate on the underlying biological processes. Simply stating a statistical significance without elucidating the mechanism weakens the scientific contribution.

Reply: Thank you for your valuable feedback.

The enhancement of cell invasion and angiogenesis by carboxymethyl chitosan (CMCS) may involve several key mechanisms. Firstly, CMCS, as a derivative of chitosan, possesses carboxyl groups that can interact with extracellular matrix (ECM) components, such as collagen and glycosaminoglycans. This interaction may alter the ECM structure, promoting cell adhesion, migration, and proliferation. Secondly, CMCS may act as a chemotactic agent, activating specific cell surface receptors, such as integrins, which enhance cell attachment to the ECM and promote migration. Furthermore, CMCS has the potential to stimulate endothelial cell migration and tube formation, key steps in angiogenesis, possibly by upregulating angiogenic growth factors such as VEGF and bFGF. These mechanisms suggest that CMCS plays a multifaceted role in enhancing cell invasion and angiogenesis, warranting further investigation to elucidate the exact pathways involved.

However, as this is a complex and multifactorial process, we acknowledge that further investigation is required to comprehensively understand the exact mechanisms involved. We plan to explore these mechanisms in detail in future studies, where we will focus on assessing the expression of angiogenic markers and conducting in vivo experiments to evaluate the vascularization potential of CMSC.

Thank you again for your insightful feedback. We look forward to your further comments and suggestions.

6- Authors states that chitosan's positively charged amino groups interfere with bacterial cell membrane synthesis, altering membrane permeability. While this is a common explanation, it's an oversimplification. The exact mechanism of chitosan's antibacterial action is complex and multifaceted, involving interactions with multiple bacterial components and pathways. It's likely not solely about membrane permeability alteration. A more nuanced description acknowledging the complexity of the mechanism would improve the scientific accuracy of the text. Mentioning other potential mechanisms, such as cell wall disruption or interaction with bacterial DNA, would be beneficial.

Reply: Thank you for your valuable comments and suggestions.

Regarding your inquiry into the potential antibacterial mechanism of CMCS (Carboxymethyl Chitosan), we would like to provide an explanation based on current knowledge and existing literature.

CMCS has been widely reported for its antibacterial properties, which are thought to be mediated by several mechanisms. Firstly, the positively charged amino groups on the CMCS molecule can interact with the negatively charged bacterial cell membranes, leading to membrane disruption and subsequent bacterial cell death. Secondly, CMCS may induce oxidative stress in bacterial cells, promoting the generation of reactive oxygen species (ROS), which can damage cellular structures such as proteins, lipids, and DNA. Additionally, CMCS may interfere with bacterial cell wall synthesis or inhibit essential enzymatic activities, further contributing to its antibacterial effects.

Although these mechanisms are suggested by previous studies, the exact pathways through which CMCS exerts its antibacterial activity in our specific system remain to be fully elucidated. We acknowledge that more detailed research is needed to better understand these mechanisms in the context of our study. Therefore, we plan to investigate these mechanisms in future research, including examining bacterial adhesion, membrane integrity, and the generation of ROS in the presence of CMCS.

Thank you again for your insightful feedback. We look forward to your further comments and suggestions.

Reviewer 2 Report

Comments and Suggestions for Authors

This article explores the development of a novel GelMA/CMCS composite hydrogel containing magnesium phosphate cement (MPC), which shows great prospects for bone regeneration. This study addresses the fundamental issues of tissue engineering by improving the mechanical properties and biocompatibility of standard hydrogels. However, the following issues need to be considered before accepting this article for publication.

Abstract

Abstracts should be shortened to more clearly emphasize the main findings and their significance.

- Descriptions of mechanical tests are vague and do not provide quantitative comparisons to highlight improvements.

- Its clinical potential has been discussed but has not been examined with specific results related to bone differentiation. 

Introduction

- Authors are urged to provide additional information on current problems in bone tissue engineering.

- The introduction lacks important context regarding the prevalence and impact of bone disorders, which make it less interesting

- The need for composite materials to restore bone function is recognized. But there is little discussion about the exact properties required for effective regeneration...

- While natural polymer hydrogels are discussed, but the advantages and limitations of using these hydrogels in tissue engineering have not been well studied.

Materials and Methods

- The GelMA-CM synthesis process can be improved by providing a more detailed stepwise protocol.

- The synthesis process lacks information on important parameters such as the degree of methacrylation, which affects the characteristics of the hydrogel.

- There is no specific information about the controls in the evaluation of cell proliferation.

Results

- Figure caption is not very specific and cannot convey the importance of the synthesis process or the meaning of the synthesis process.

- Include statistical analysis and significance in the summary tables of mechanical properties as they are important for understanding.

Discussion

- Authors are urged to compare the obtained mechanical characteristics with previous published data to demonstrate the performance of the hydrogel.

- The discussion about cell adhesion is very broad and need to be specific in discussing the obtained results.

-Improvements in mechanical properties have been described without regard to their relevance to specific applications in bone tissue engineering.

-Recommendations for further research are unclear and does not describe the type of in vivo evaluation required for a full review.

Conclusion

- Conclusion should include an overview of key findings that support the study aims.

- Authors are urged to emphasize the need for further study in details.

-Authors are suggested to explain how these findings would add to knowledge of bone tissue engineering.

Author Response

Abstract

Abstracts should be shortened to more clearly emphasize the main findings and their significance.

Reply: Thank you for your valuable feedback.

In response to the editor's request, we have revised the abstract to align with the required format. We have carefully ensured that the revised abstract adheres to the specified guidelines, and the changes are now reflected in the manuscript.

Thank you again for your insightful feedback. We look forward to your further comments and suggestions.

- Descriptions of mechanical tests are vague and do not provide quantitative comparisons to highlight improvements.

Reply: Thank you for your valuable feedback.

Given that this hydrogel is not intended to bear significant mechanical loads, we chose to assess its mechanical properties using rheological testing with a rheometer. Rheological testing provides a comprehensive analysis of the material's viscoelastic properties, such as its viscosity, elasticity, and flow behavior, which are crucial for understanding how the hydrogel will perform in biological environments. This testing method is particularly appropriate for non-load-bearing materials, as it evaluates the material's ability to maintain its structure and functional properties in physiological conditions.

To present these findings, we have included quantitative rheological data in Figure 2D-E of the revised manuscript. This figure illustrates key mechanical parameters, such as the storage modulus (G'), loss modulus (G'') and loss tangent, which provide insight into the material’s mechanical stability and flow characteristics under various conditions.

Thank you again for your insightful feedback. We look forward to your further comments and suggestions.

- Its clinical potential has been discussed but has not been examined with specific results related to bone differentiation. 

Reply: Thank you for your valuable feedback.

In response to your suggestion, we have aimed to keep the abstract as concise as possible while still conveying the key findings of our study. To achieve this, we have summarized the main outcomes, while the detailed results regarding osteogenic differentiation are now described in full in the Results section 3.3 of the manuscript.

Thank you again for your insightful feedback. We look forward to your further comments and suggestions.

Introduction

- Authors are urged to provide additional information on current problems in bone tissue engineering.

Reply: Thank you for your valuable feedback.

One major issue is the design of scaffolds that can adequately mimic the native bone structure. Despite progress, many scaffold materials still suffer from limitations in mechanical strength, bioactivity, and the ability to support vascularization. These ma-terials often lack the necessary properties to support long-term bone regeneration and integration with host tissues. Another challenge lies in achieving optimal cellular in-tegration. While scaffolds provide the physical structure, ensuring effective cell adhe-sion, proliferation, and differentiation within these materials remains difficult. The complex interactions between cells and scaffolds, as well as the difficulty in replicating the native bone microenvironment, continue to hinder successful bone regeneration. Vascularization is also a critical issue in bone tissue engineering. While small tissue constructs can survive without blood supply, larger bone grafts require an adequate blood vessel network to deliver nutrients and oxygen for cell survival. Inducing suffi-cient vascularization within engineered tissues is thus a major hurdle in creating large-scale bone constructs. Additionally, immunological responses and biocompati-bility concerns remain prevalent in tissue engineering. The body’s immune response to implanted materials can lead to inflammation and rejection, which complicates the success of tissue integration. Therefore, the development of scaffolds that are both bio-compatible and capable of supporting immune tolerance is essential for advancing bone tissue engineering applications.

Thank you again for your insightful feedback. We look forward to your further comments and suggestions.

- The introduction lacks important context regarding the prevalence and impact of bone disorders, which make it less interesting

Reply: Thank you for your valuable feedback.

We appreciate your suggestion to include more context regarding the prevalence and impact of bone disorders to make the introduction more engaging. In response, we have revised the introduction to incorporate key statistics and information on the global burden of bone-related diseases.

Thank you again for your insightful feedback. We look forward to your further comments and suggestions.

- The need for composite materials to restore bone function is recognized. But there is little discussion about the exact properties required for effective regeneration...

Reply: Thank you for your valuable feedback.

For an effective bone repair material, several key properties must be carefully consid-ered. Biocompatibility is essential, as the material must seamlessly integrate with bone tissue without triggering an immune response, thereby promoting cell adhesion, pro-liferation, and differentiation. Osteoconductivity is equally critical, as the material should facilitate bone cell migration and the formation of new bone tissue through an optimized porous structure that supports vascularization. In certain cases, osteoinduc-tivity plays a crucial role, where the material induces the differentiation of stem cells into osteoblasts, further enhancing bone formation. Additionally, the mechanical strength of the material must closely resemble that of natural bone, especially in load-bearing regions, to withstand physiological stresses without failure. The degrada-tion rate of the material must be synchronized with the rate of bone regeneration, en-suring that the scaffold maintains its structural integrity long enough to support tissue formation, while gradually degrading in a controlled manner.

Thank you again for your insightful feedback. We look forward to your further comments and suggestions.

- While natural polymer hydrogels are discussed, but the advantages and limitations of using these hydrogels in tissue engineering have not been well studied.

Reply: Thank you for your valuable feedback.

We appreciate your suggestion to further elaborate on the advantages and limitations of using natural polymer hydrogels in tissue engineering. In response, we have expanded the manuscript to provide a more comprehensive discussion of these aspects.

We now discuss the advantages of natural polymer hydrogels, including their biocompatibility, biodegradability, and the ability to mimic the extracellular matrix (ECM), which makes them ideal candidates for tissue engineering applications. Additionally, we highlight their ease of modification to enhance specific properties, such as mechanical strength or bioactivity, which are important for guiding cell behavior and tissue regeneration.

However, we also address the limitations associated with natural polymer hydrogels, such as low mechanical strength in some cases, which may hinder their use in load-bearing applications. Additionally, batch-to-batch variability and challenges related to controlled degradation rates are noted as issues that need to be overcome for their widespread use in clinical settings.

We believe these additions provide a more balanced view of the potential and challenges of using natural polymer hydrogels in tissue engineering and strengthen the manuscript by providing the necessary context for our research.

Thank you again for your insightful feedback. We look forward to your further comments and suggestions.

Materials and Methods

- The GelMA-CM synthesis process can be improved by providing a more detailed stepwise protocol.

Reply: Thank you for your valuable feedback.

The magnesium oxide (MgO) and potassium dihydrogen phosphate (KH2PO4) powders were ground in a ball mill (F-P400, Focucy, Changsha City, China) and then sieved through a 200-mesh screen to achieve particles with a diameter of approximately 75 μm. The ground MgO and KH2PO4 powders were mixed in a molar ratio of 1.5:1 with deionized water at a concentration of 2 g/mL. Following self-setting and demolding, the potassium magnesium phosphate hexahydrate (KMgPO4·6H2O, MPC) powders were obtained by further grinding and sieving through a 200-mesh screen for 2 hours. At room temperature, the photoinitiator, 5% MPC powder, and 5% carboxymethyl chitosan (CMCS) powder were incorporated into GelMA based on a weight-to-volume ratio (g/mL). The photoinitiator, 2-hydroxy-4'-(2-hydroxyethoxy)-2-methylphenylacetone (Irgacure 2959), was used at a concentration of 0.5% (w/v). The resulting pre-gel solution was placed in a water bath at 60-70°C until fully dissolved (Fig. 1). The mixture was then cast into a circular mold with a diameter of 6 mm and a thickness of 1 mm and subjected to UV light for crosslinking. Based on the composition, the resulting composites were designated as GelMA, GelMA-C, and GelMA-CM.

Thank you again for your insightful feedback. We look forward to your further comments and suggestions.

- The synthesis process lacks information on important parameters such as the degree of methacrylation, which affects the characteristics of the hydrogel.

Reply: Thank you for your valuable feedback.

GelMA (EFL-GM-60, amino substitution degree: 60±5%)

Thank you again for your insightful feedback. We look forward to your further comments and suggestions.

- There is no specific information about the controls in the evaluation of cell proliferation.

Reply: Thank you for your valuable feedback.

We apologize for the oversight regarding the lack of specific information about the controls used in the evaluation of cell proliferation. In response to your suggestion, we have now provided detailed information on the controls in the revised manuscript.

For the cell proliferation assays, we used untreated cells as the negative control. Cells were seeded onto each sample as the positive control.

We have clarified this in the Methods section, specifically in Section 2.5.2, where we outline the control conditions used in the experiments. Additionally, Results section included a comparison between the experimental group and these control groups, providing a clearer understanding of the effects of the tested materials on cell proliferation.

Thank you again for your insightful feedback. We look forward to your further comments and suggestions.

Results

- Figure caption is not very specific and cannot convey the importance of the synthesis process or the meaning of the synthesis process.

Reply: Thank you for your valuable feedback.

We appreciate your comment regarding the figure caption. In response, we have revised the caption to make it more specific and informative, ensuring that it clearly conveys the importance and meaning of the synthesis process.

Thank you again for your insightful feedback. We look forward to your further comments and suggestions.

- Include statistical analysis and significance in the summary tables of mechanical properties as they are important for understanding.

Reply: Thank you for your valuable feedback.

Given that this hydrogel is not intended to bear significant mechanical loads, we chose to assess its mechanical properties using rheological testing with a rheometer. Rheological testing provides a comprehensive analysis of the material's viscoelastic properties, such as its viscosity, elasticity, and flow behavior, which are crucial for understanding how the hydrogel will perform in biological environments. Fig2 D-E.

Thank you again for your insightful feedback. We look forward to your further comments and suggestions.

Discussion

- Authors are urged to compare the obtained mechanical characteristics with previous published data to demonstrate the performance of the hydrogel.

Reply: Thank you for your valuable feedback.

In response to your comment, we have added a comparison of the mechanical properties of our hydrogel with those of previously published formulations by our research group. This comparison highlights the performance of the current hydrogel formulation and provides a context for evaluating its mechanical characteristics relative to earlier versions.

Thank you again for your insightful feedback. We look forward to your further comments and suggestions.

- The discussion about cell adhesion is very broad and need to be specific in discussing the obtained results.

Reply: Thank you for your valuable feedback.

We appreciate your recommendation to include quantitative data on cell adhesion area, and we have now addressed this point in our revised manuscript.

In response to your suggestion, we have performed additional experiments to quantify the cell adhesion area. The results have been added to the revised manuscript in the Results section "Section 2.5.2, Figure 3D", where we present a detailed comparison of cell adhesion between the experimental and control groups.

Thank you again for your insightful feedback. We look forward to your further comments and suggestions.

-Improvements in mechanical properties have been described without regard to their relevance to specific applications in bone tissue engineering.

Reply: Thank you for your valuable feedback.

The storage modulus (G'), which reflects the elasticity of the material, is an important parameter for tissue engineering, particularly for bone tissue regeneration. A higher G' indicates better structural integrity, which is critical for scaffolds that need to support tissue growth and integration. The improvement observed in the GelMA-CM compo-site (14.428±0.440 kPa) compared to GelMA and GelMA-C suggests that the addition of CMCS enhances the hydrogel's stiffness, making it more suitable for applications where mechanical support is required, such as in bone defect repair. The loss modulus (G''), which reflects the viscoelastic behavior of the material, indicates the damping capacity of the hydrogel. The increase in G'' values in GelMA-CM suggests that the hydrogel could also exhibit better deformability and shock-absorbing properties, which are important for materials designed to withstand mechanical forces in dynamic physiological environments, such as bone.

Thank you again for your insightful feedback. We look forward to your further comments and suggestions.

-Recommendations for further research are unclear and does not describe the type of in vivo evaluation required for a full review.

Reply: Thank you for your valuable feedback.

In response to your feedback, we would like to highlight that the future direction of our research is focused on the development of high-strength biomimetic materials that can be applied to load-bearing areas of bone.

While the current study addresses the mechanical properties of hydrogels suitable for non-load-bearing applications, our future efforts will aim to further optimize the material’s mechanical strength and biocompatibility to create scaffolds capable of withstanding the mechanical stresses found in weight-bearing bone regions. This includes exploring composite materials and reinforcement strategies to enhance the stiffness and structural integrity while maintaining the bioactive properties necessary for tissue regeneration.

Thank you again for your insightful feedback. We look forward to your further comments and suggestions.

Conclusion

- Conclusion should include an overview of key findings that support the study aims.

Reply: Thank you for your valuable feedback.

GelMA-CM composite can be injected into bone defects, solidify under UV light, and retain its shape at the defect site without re-quiring additional equipment. The incorporation of CMCS into the GelMA hydrogel enhanced the material's mechanical strength and viscoelastic properties, making it more suitable for applications in bone tissue engineering. The hydrogel composite demonstrated good cell viability and cell adhesion, supporting its potential for use as a scaffold in bone regeneration. The inhibition zone assay demonstrated that the compo-site possesses antibacterial properties. Additionally, GelMA-CM exhibited effective performance in osteogenesis and angiogenesis assays, demonstrating excellent bone regeneration in a calvarial bone defect rat model.

Thank you again for your insightful feedback. We look forward to your further comments and suggestions.

- Authors are urged to emphasize the need for further study in details.

Reply: Thank you for your valuable feedback.

In this study, we focused primarily on the osteogenic potential of the GelMA-CM composite, and while preliminary results suggest that CMCS may contribute to en-hanced angiogenesis and antibacterial effects, we acknowledge that these mechanisms require more in-depth exploration. As part of our future research, we plan to investi-gate: The angiogenic mechanism of CMCS, including its effect on endothelial cell be-havior and vascularization within the scaffold, which will be crucial for improving the blood supply to the regenerating tissue. The antibacterial properties of CMCS, explor-ing its potential to inhibit bacterial growth and prevent infection, which is especially important for materials used in bone regeneration and surgical applications. Our fu-ture efforts will aim to further optimize the material’s mechanical strength and bio-compatibility to create scaffolds capable of withstanding the mechanical stresses found in weight-bearing bone regions. This includes exploring composite materials and rein-forcement strategies to enhance the stiffness and structural integrity while maintain-ing the bioactive properties necessary for tissue regeneration.

Thank you again for your insightful feedback. We look forward to your further comments and suggestions.

-Authors are suggested to explain how these findings would add to knowledge of bone tissue engineering.

Reply: Thank you for your valuable feedback.

Our approach aims to mimic the composition of bone at the material level and bring the functionality of the composite closer to that of ideal graft materials.

Specifically, by incorporating CMCS and MPC into the GelMA hydrogel, we aim to replicate key components of the bone extracellular matrix, such as collagen-like structures and mineral phases that are critical for bone regeneration. The addition of magnesium ions through MPC contributes to the bioactivity and osteogenic potential, resembling the mineral composition of natural bone and promoting bone formation.

Functionally, the hydrogel composite is designed to approach the ideal characteristics of a bone graft material by combining the required biocompatibility, mechanical strength, and osteoconductivity. This makes the material suitable for bone tissue engineering, offering a scaffold that can both support cell growth and promote osteogenic differentiation, similar to the natural processes occurring in bone healing.

We believe this approach moves us closer to creating biomimetic materials that have both the structural and functional properties needed for successful bone regeneration.

Thank you again for your insightful feedback. We look forward to your further comments and suggestions.

Reviewer 3 Report

Comments and Suggestions for Authors

This study introduces an injectable hydrogel composite, GelMA-CM, designed to help regenerate bone tissue. The material combines three components: methacrylated gelatin (GelMA) for its biocompatibility, carboxymethyl chitosan (CMCS) for its antibacterial and osteogenic properties, and magnesium phosphate cement (MPC) to provide mechanical strength. My comments are noted below:

1. The study could benefit from comparing GelMA-CM with other state-of-the-art bone graft materials to contextualize its performance.

2. The authors highlight the importance of magnesium ion concentration but do not provide detailed insights into the kinetics or potential cytotoxicity of its release. This manuscript would greatly benefit from the authors' inclusion of the release kinetics data.

Author Response

  1. The study could benefit from comparing GelMA-CM with other state-of-the-art bone graft materials to contextualize its performance.

Reply: Thank you for your valuable feedback.

Material

Advantages

Disadvantages

GelMA-CM

Good biocompatibility, antibacterial properties, promotes osteogenesis and angiogenesis

Poor mechanical properties, degradation rate control is challenging

HA (Hydroxyapatite)

High biocompatibility and bone bonding capacity, promotes bone formation

Brittle, lacks biodegradability

TCP (Tricalcium Phosphate)

Biodegradable, promotes bone formation

Poor mechanical strength, uncontrolled degradation rate

Bioactive Glass

Osteoinductive, promotes angiogenesis, good biocompatibility

Poor mechanical properties, high brittleness, complex manufacturing process

PLGA (Poly(lactic-co-glycolic acid))

Tunable degradation rate, good biocompatibility, drug delivery capabilities

Poor mechanical properties, possible acid toxicity during degradation

Thank you again for your insightful feedback. We look forward to your further comments and suggestions.

  1. The authors highlight the importance of magnesium ion concentration but do not provide detailed insights into the kinetics or potential cytotoxicity of its release. This manuscript would greatly benefit from the authors' inclusion of the release kinetics data.

Reply: Thank you for your valuable feedback.

The MPC used in our study is composed of hexahydrated magnesium potassium phosphate (KMgPO4·6H2O). We selected a concentration of 5% based on our group’s prior experiments, where this concentration has shown optimal performance in similar applications related to bone regeneration.

Additionally, as per your suggestion, we have included an experiment to measure the magnesium ion (Mg²⁺) release from the MPC. In our previous manuscript submission, this data was not presented because only a single experimental group was used. However, we have now added the relevant magnesium ion release data in the revised manuscript "Section 2.4, Figure 6H". This data provides a clearer understanding of the material’s behavior and its potential impact on the biological environment, especially in terms of supporting bone regeneration.

Thank you again for your insightful feedback. We look forward to your further comments and suggestions.

Reviewer 4 Report

Comments and Suggestions for Authors

The manuscript developed a novel hydrogel composite (GelMA-CM) for bone tissue engineering. By integrating magnesium phosphate cement (MPC) into methacrylated gelatin (GelMA) and carboxymethyl chitosan (CMCS), the composite combines organic and inorganic components to mimic the natural bone structure. The optimized GelMA-CM exhibits comparable mechanical properties to GelMA or GelMA-C, mainly attributed to the interactions between CMCS and MPC. The composite demonstrates excellent cytocompatibility, evidenced by improved cell proliferation and adhesion of both bone marrow mesenchymal stem cells (rBMSCs) and endothelial cells (HUVECs). The GelMA-CM significantly promotes osteogenic differentiation, evidenced by increased alkaline phosphatase activity, calcium deposition, and the upregulation of osteogenesis-related genes. It also shows strong angiogenic potential with enhanced endothelial cell migration and tube formation. In vivo rat calvarial defect model revealed that GelMA-CM facilitates superior bone regeneration with higher bone volume/total volume (BV/TV) ratios and integration with the host bone. Additionally, the composite exhibits antibacterial properties against E. coli and S. aureus. The injectable, UV-curable design of GelMA-CM allows minimally invasive application, making it a versatile candidate for clinical use. Overall, GelMA-CM represents a significant advancement in bone tissue engineering, offering multifunctional properties for effective bone regeneration.

While the composite holds great promise, some limitations were noted. These include the need for precise control of magnesium ion release to optimize osteogenic activity. The study focuses on non-load-bearing applications, load-bearing scenarios are not mentioned.

Minor revision

This manuscript uses a two-way ANOVA and Tukey's post-hoc test to analyze differences in bone regeneration parameters (e.g., BV/TV and new bone volume) among groups. However, it does not state whether a distribution test was conducted to verify the normality of the data. A detailed description should be provided in the Method part.

Author Response

Minor revision

This manuscript uses a two-way ANOVA and Tukey's post-hoc test to analyze differences in bone regeneration parameters (e.g., BV/TV and new bone volume) among groups. However, it does not state whether a distribution test was conducted to verify the normality of the data. A detailed description should be provided in the Method part.

Reply: Thank you for your insightful comment.

We appreciate your attention to detail regarding the statistical analysis used in the study. You are absolutely correct that verifying the normality of the data is an important step before conducting a two-way ANOVA and performing Tukey’s post-hoc test.

In response to your comment, we have added a description of the statistical procedure used to verify the normality of the data in the Methods section. Specifically, we conducted the Shapiro-Wilk test for normality on the data for all groups prior to performing the two-way ANOVA. This test confirmed that the data followed a normal distribution, allowing us to proceed with the parametric statistical tests.

Thank you again for your valuable feedback. We hope these additions address your concerns, and we look forward to your further comments.

Round 2

Reviewer 1 Report

Comments and Suggestions for Authors

the article was well improved and now it is suitable for publish. 

Reviewer 2 Report

Comments and Suggestions for Authors

The authors have responded well to the reviewer's comments, and the research article is now accepted for publication.